# The metabolomic profile associated with clustering of cardiovascular risk factors—A multi-sample evaluation

Lars Lind[1]*, Johan Sundström[1], Sölve Elmståhl[2], Koen F. Dekkers[1], J. Gustav Smith[3,4,5,6], Gunnar Engström[2], Tove Fall[1], Johan Ärnlöv[7,8]

**1** Department of Medical Sciences, Uppsala University, Uppsala, Sweden, **2** Department of Clinical Sciences, Lund University, Malmö, Sweden, **3** Department of Cardiology, Clinical Sciences, Lund University and Skåne University Hospital, Lund, Sweden, **4** Wallenberg Laboratory, Department of Molecular and Clinical Medicine, Institute of Medicine, Gothenburg University, Gothenburg, Sweden, **5** Department of Cardiology, Sahlgrenska University Hospital, Gothenburg, Sweden, **6** Wallenberg Center for Molecular Medicine and Lund University Diabetes Center, Lund University, Lund, Sweden, **7** Division of Family Medicine and Primary Care, Department of Neurobiology, Care Science and Society, Karolinska Institutet, Huddinge, Sweden, **8** School of Health and Social Studies, Dalarna University, Falun, Sweden

* lars.lind@medsci.uu.se

**Data Availability Statement:** Due to Swedish laws on personal integrity and health data, as well as the decision by the Ethics Committee, we are not

## Abstract

### Background

A clustering of cardiovascular risk factors is denoted the metabolic syndrome (MetS), but the mechanistic underpinnings of this clustering is not clear. Using large-scale metabolomics, we aimed to find a metabolic profile common for all five components of MetS.

### Methods and findings

791 annotated non-xenobiotic metabolites were measured by ultra-performance liquid chromatography tandem mass spectrometry in five different population-based samples (Discovery samples: EpiHealth, n = 2342 and SCAPIS-Uppsala, n = 4985. Replication sample: SCAPIS-Malmö, n = 3978, Characterization samples: PIVUS, n = 604 and POEM, n = 501). MetS was defined by the NCEP/consensus criteria. Fifteen metabolites were related to all five components of MetS (blood pressure, waist circumference, glucose, HDL-cholesterol and triglycerides) at a false discovery rate of <0.05 with adjustments for BMI and several life-style factors. They represented different metabolic classes, such as amino acids, simple carbohydrates, androgenic steroids, corticosteroids, co-factors and vitamins, ceramides, carnitines, fatty acids, phospholipids and metabolonic lactone sulfate. All 15 metabolites were related to insulin sensitivity (Matsuda index) in POEM, but only Palmitoyl-oleoyl-GPE (16:0/18:1), a glycerophospholipid, was related to incident cardiovascular disease over 8.6 years follow-up in the EpiHealth sample following adjustment for cardiovascular risk factors (HR 1.32 for a SD change, 95%CI 1.07–1.63).

### Conclusion

A complex metabolic profile was related to all cardiovascular risk factors included in MetS independently of BMI. This profile was also related to insulin sensitivity, which provide

allowed to make any data including health variables open to the public, even if made anonymous. The data could be shared with other researchers after a request to the head of the department (eva.lindberg@medsci.uu.se).

**Funding:** JGS was supported by grants from the Swedish Heart-Lung Foundation (2019-0526), the Swedish Research Council (2017-02554), the European Research Council (ERC-STG-2015-679242), Skåne University Hospital, governmental funding of clinical research within the Swedish National Health Service, a generous donation from the Knut and Alice Wallenberg foundation to the Wallenberg Center for Molecular Medicine in Lund, and funding from the Swedish Research Council (Linnaeus grant Dnr 349-2006-237, Strategic Research Area Exodiab Dnr 2009-1039) and Swedish Foundation for Strategic Research (Dnr IRC15-0067) to the Lund University Diabetes Center. GE was supported by grants from the Swedish Heart-Lung Foundation (20200173), Swedish Research Council (2019-01236) TF was supported by grants from Swedish Research Council (2019-01471), the Swedish Heart-Lung Foundation (2019-0505), and the European Research Council (ERC-2018-STG 801965). JÄ was supported by grants from Swedish research council (2019-01015 and 2020-00243) and the Swedish Heart Lung foundation (20180343). The main funding body of The Swedish CArdioPulmonary bioImage Study (SCAPIS) is the Swedish Heart-Lung Foundation. The study is also funded by the Knut and Alice Wallenberg Foundation, the Swedish Research Council, and VINNOVA (Sweden's Innovation agency), the University of Gothenburg and Sahlgrenska University Hospital, Karolinska Institutet and Stockholm county council, Linköping University and University Hospital, Lund University and Skåne University Hospital, Umeå University and University Hospital, Uppsala University and University Hospital. The EpiHealth study is funded as a strategic research area (SFO) by the Swedish government. The data handling were enabled by resources in project sens2019512 provided by the Swedish National Infrastructure for Computing (SNIC) at Uppsala Multidisciplinary Center for Advanced Computational Science (UPPMAX), partially funded by the Swedish Research Council through grant agreement no. 2018-05973. The funders had no role in study design, data collection and analysis, decision to publish, or preparation of the manuscript.

**Competing interests:** The authors have declared that no competing interests exist.

further support for the importance of insulin sensitivity as an important underlying mechanism in the clustering of cardiovascular risk factors.

## Introduction

In 1988, Reaven and others suggested the existence of a metabolic syndrome (MetS), a clustering of cardiovascular risk factors [1, 2] in certain individuals, with insulin resistance as a common denominator. However, the molecular mechanisms behind the clustering of cardiovascular risk factors seen in some individuals are not clear.

One way to disclose novel mechanisms is to use genomics to search for genetic loci being in common for the components included in the most frequently used definition of the syndrome (NCEP/consensus criteria; high blood pressure, increased waist circumference, high fasting glucose, low HDL-cholesterol and increased triglycerides.). Using this genomic approach, three loci were related to all five MetS components in one study (nearest genes *LINC0112*, *C5orf67*, and *GIP*) [3].

Another way is to use proteomics in a similar fashion. In such attempt using targeted proteomics [4], we found 20 proteins being related to all five MetS components, representing several pathophysiological pathways. (immunomodulation at different level, regulation of adipocyte differentiation, lipid, carbohydrate, and amino acid metabolism; or insulin-like growth factor signaling).

A third -omics technology frequently used nowadays is metabolomics, including small compounds with a molecular weight <1.5 kD. Metabolomics have extensively been used to characterize the metabolic landscape of obesity and diabetes [5, 6]. Moreover, several studies have also been published on metabolomics in MetS [7]. Most of those studies have evaluated individuals with MetS vs controls, while no study has reported the metabolic fingerprint being related to all risk factors in the syndrome. Thus, despite these previous studies there is still a knowledge gap in that respect. We hypothesized that an identification of the metabolic fingerprint being related to all risk factors in the syndrome could shed a light on the underlying mechanisms behind the clustering of cardiovascular risk factors.

It is well known that MetS is related to future atherosclerotic cardiovascular disease (CVD) events [8, 9], although the increased risk by having MetS is not greater than sum of the risk factors included in the syndrome [10]. It would therefore be of interest to investigate if the metabolic fingerprint being related to all risk factors in the syndrome also is related to future atherosclerotic events.

In the present study, the primary aim was to use large-scale metabolomics data to identify metabolites common to all cardiovascular risk factors included in MetS. For this task, we used data from three different cohorts with together >11,000 individuals using a discovery/validation approach. As a secondary aim, we further evaluated if the identified metabolic profile being related to all five MetS components was associated with prevalent MetS in another two cohorts, as well as with insulin resistance. A third aim was to investigate if metabolites being related to all risk factors in the syndrome also was related to future atherosclerotic events.

## Material and methods

### Study samples

Four different population-based samples were used in the present study. All studies were conducted in accordance with the principles of the Declaration of Helsinki, approved by the

responsible ethics committees, and written informed consent was obtained from all participants (Etikprövningsmyndigheten; Dnr 2021–00134).

## SCAPIS (Swedish CArdioPulmonary imaging study)

As a collaboration project between six Swedish universities, a total of 30,000 men and women aged 50–65 years have been investigated in six Swedish cities [11].

Besides quantification of traditional cardiovascular risk factors, an extensive imaging program has been performed, including computed tomography (CT) coronary angiography, and carotid artery atherosclerosis with ultrasound. Metabolomic measurements have been performed in 4985 individuals collected in Uppsala and in 3978 individuals collected in Malmö. The data collected at the two sites were treated as separate samples in our statistical analyses.

## EpiHealth

Starting April 27th 2011, men and women in the age groups 45 to 75 in two Swedish towns, Uppsala and Malmö, have been invited in a random fashion to a health screening survey, called EpiHealth [12]. In 2018, data on approximately 25,000 individuals had been collected. Traditional CV risk factors and fat mass (bioimpedance) have been recorded. Metabolomic data have been collected in a subsample of subjects attending the Uppsala part of the study (n = 2342). This sample has been followed for almost 10 years regarding incident CVD.

## POEM (Prospective investigation of obesity, energy and metabolism)

The population-based POEM study was conducted in men and women living in Uppsala, Sweden all aged 50 years [13]. Between Oct 2010 and Oct 2016, 502 individuals were investigated. A 2h OGTT with insulin determinations each 30th min was performed for calculations of insulin sensitivity. Metabolomics measurements have been performed in all participants, but one (due to missing plasma).

## PIVUS (Prospective investigation of the vasculature in Uppsala seniors)

The population-based PIVUS study was conducted in men and women living in Uppsala, Sweden all aged 70 years [14]. Between April 2001 and June 2004, 1016 individuals were investigated. The investigation was repeated at ages 75 and 80 years. At the age of 80, metabolomics measurements were performed in the total sample (n = 604).

In all samples, but EpiHealth, fasting blood samples were collected in the morning after an overnight fast. In EpiHealth, blood samples were collected after 6 hours of fasting. Blood lipids and glucose were measured locally at the hospitals in Uppsala and Malmö, respectively. Blood pressure was measured twice in sitting position and the mean value was used. Waist circumference was measured at the umbilical level. BMI was calculated from height and weight measurements.

The metabolic syndrome (MetS) was determined using the NCEP-based consensus criteria [15]. The five components were defined as follows: Blood pressure $\geq$ 130/85 mmHg or antihypertensive treatment, fasting plasma glucose $\geq$ 6.1 mmol/l or antidiabetic treatment, serum triglycerides $\geq$ 1.7 mmol/l, waist circumference > 102 cm in men and > 88 cm in women, HDL-cholesterol < 1.0 mmol/l in men and < 1.3 in women. Three of the mentioned five criteria should be fulfilled for MetS.

Alcohol intake, smoking habits, exercise habits and education level were assessed by questionnaires.

In the POEM study only, an 2h oral glucose tolerance test (OGTT, 75g glucose) was carried out and glucose and insulin were measured at times 0, 30, 60, 90, and 120 min. From these data, insulin sensitivity was assessed by the Matsuda index [16].

## Metabolomics

In all samples, non-targeted metabolomics (Metabolon inc., USA) was performed on plasma samples being stored at -80˚ C. Samples were prepared using the automated MicroLab STAR® system from Hamilton Company. Several recovery standards were added prior to the first step in the extraction process for quality control purposes. To remove protein, dissociate small molecules bound to protein or trapped in the precipitated protein matrix, and to recover chemically diverse metabolites, proteins were precipitated with methanol under vigorous shaking for 2 min (Glen Mills GenoGrinder 2000) followed by centrifugation. The resulting extract was divided into five fractions: two for analysis by two separate reverse phases (RP)/ UPLC-MS/MS methods with positive ion mode electrospray ionization (ESI), one for analysis by RP/UPLC-MS/MS with negative ion mode ESI, one for analysis by HILIC/UPLC-MS/MS with negative ion mode ESI, and one sample was reserved for backup. Only annotated, non-xenobiotic metabolites with a call rate >75% in were used in the analyses (n = 791). The values were normalized and given in arbitrary units.

## Atherosclerotic CVD definition

Using data from the Swedish cause of death and in-hospital care registers, we defined a combined end-point for atherosclerotic CVD being either fatal or non-fatal acute myocardial infarction or ischemic stroke (ICD-10 codes I20 or I63-I66). Incident cases of atherosclerotic CVD were only investigated in the EpiHealth sample, since the other samples had yet too short follow-up period. The median follow-up period in EpiHealth was 8.6 years. The censor date of the follow-up was Dec 31, 2020.

## Statistics

All metabolomic data were subjected to inverse rank normalization within each sample.

One linear regression model for each metabolite was performed vs the five MetS components separately (systolic blood pressure (SBP), fasting glucose (GLU), triglycerides (TG), HDL-cholesterol (HDL) and waist circumference (WC)). All of these analyses were adjusted for age, sex and BMI, as well as for the life-style factors alcohol intake, smoking habits, exercise habits and education level. These analyses were carried out separately in SCAPIS-Uppsala, SCAPIS-Malmö and EpiHealth.

We thereafter undertook a discovery/validation approach with an inverse-variant weighted fixed effect meta-analysis of the two Uppsala samples (SCAPIS-Uppsala and EpiHealth) as the discovery step and SCAPIS-Malmö as the validation step. In both steps, we required the metabolites to show a false discovery rate (FDR) <0.05 in order to be judged as a validated metabolite.

The 15 validated metabolites being related to all 5 MetS criteria and with the same sign of the beta coefficient for all risk factors (except for HDL), were one by one related to incident atherosclerotic CVD in the EpiHealth sample using Cox proportional hazard analysis. Two set of adjustments were performed. First, adjustment for age and sex. Second, adjustment also for the classical risk factors systolic blood pressure, smoking (current yes/no), HDL and LDL-cholesterol, BMI and diabetes. FDR<0.05 for the age and sex-adjusted analysis in combination with p<0.05 for the multiple adjustment was regarded as significant in this analysis. Subjects with prevalent CVD at baseline were excluded from this analysis.

The 15 validated metabolites being related to all 5 MetS criteria and with the same sign of the beta coefficient for all risk factors (except for HDL), were one by one related to prevalent MetS in POEM and PIVUS by use of logistic regression analysis adjusting for age, sex and BMI, as well as for the life-style factors smoking habits, exercise habits and education level (no information on alcohol intake). The results from these two studies were then meta-analyzed (inverse-variant weighted fixed effect) and FDR<0.05 was considered as significant.

The 15 validated metabolites being related to all 5 MetS criteria were further investigated for association with insulin resistance (Matsuda index, ln-transformed) in POEM. Linear regression analysis adjusting for age, sex and BMI, as well as for the life-style factors smoking habits, exercise habits and education level (no information on alcohol intake) was carried out for each metabolite. FDR<0.05 was considered as significant.

For the 15 validated metabolites being related to all five MetS components, we searched for genetic associations at http://metabolomics.helmholtz-muenchen.de/gwas/ using the Shin et al. analysis [17] and in the GWAS catalogue: https://www.ebi.ac.uk/gwas/.

We used the Mendelian randomization framework to evaluate shared genetics for these metabolites and genetics for MetS using an already published GWAS for MetS [18]. The Wald ratio was used to obtain the causal estimate when only one locus linked to a metabolite (mQTL) was used as instrumental variable. An inverse-variance weighted fixed-effect meta-analysis (IVW) was used when more than one SNP was used as instrumental variable. Only SNPs with $p<5^*10^{-8}$ and not in LD (<0.001) were used in this evaluation. The genetics for metabolites were downloaded from the KORA Helmholtz Zentrum Munich metabolomics GWAS server (http://metabolomics.helmholtz-muenchen.de/gwas/) [13].

STATA16 (Stat inc, College Station, TX) was used for calculations.

## Results

Baseline characteristics of the four cohorts are provided in Table 1.

Of the 791 metabolites, 135 metabolites were identified and validated to be associated with SBP, 488 with HDL, 512 with triglycerides, 404 with glucose and 252 were identified to be associated with waist circumference. All analyses were adjusted for age, sex, exercise habits, alcohol intake, education level, smoking and BMI. See overview of the metabolite/MetS components associations in S1 Table.

Fifteen metabolites were associated with all five MetS components (see Fig 1 for overview and S2 Table for details). They represented different metabolic classes, such as amino acids (glycine, S-methylcysteine sulfoxide), simple carbohydrates (glucose, glycerate, lactate), androgenic steroids (11beta-hydroxyandrosterone glucuronide), corticosteroids (cortolone glucuronide, tetrahydrocortisol glucuronide), co-factors and vitamins (oxalate (ethanedioate), carotene diol), ceramides (N-stearoyl-sphinganine (d18:0/18:0)), carnitines (pimeloylcarnitine/3-methyladipoylcarnitine (C7-DC)), fatty acids (hydroxy-CMPF), phospholipids (1-palmitoyl-2-oleoyl-phosphatidylethanolamine (GPE) (16:0/18:1)), and metabolonic lactone sulfate.

Nine of those metabolites showed FDR<0.05 vs prevalent MetS in POEM and PIVUS (Table 2).

All 15 metabolites were associated with (FDR<0.05) the Matsuda index in POEM.

In the Epihealth cohort, 98 subjects experienced an atherosclerotic CVD event (myocardial infarction or ischemic stroke) during a median follow-up period of 8.6 years (maximal 9.6 years, 18,922 person years at risk). Two of the 15 metabolites being associated with all five MetS components showed FDR<0.05 in the age and sex-adjusted analysis, but only one metabolite, 1-palmitoyl-2-oleoyl-GPE (16:0/18:1), also showed p<0.05 following adjustment

**Table 1. Basic characteristics of the samples.** Means and (SD) are given, or proportions in %.

| | SCAPIS-Uppsala | SCAPIS-Malmö | EpiHealth | POEM | PIVUS |
|---|---|---|---|---|---|
| **n** | 4985 | 3978 | 2342 | 502 | 604 |
| **Age** | 57.6 (4.4) | 57.4 (4.2) | 61.1 (8.4) | 50 (0.1) | 80 (0.2) |
| **Female sex** | 51% | 52% | 50% | 50% | 50% |
| **Systolic blood pressure (mmHg)** | 125 (16) | 122 (16) | 135 (17) | 126 (16) | 147 (19) |
| **Diastolic blood pressure (mmHg)** | 77 (10) | 75 (10) | 83 (9) | 77 (10) | 74 (9) |
| **HDL-cholesterol (mmol/l)** | 1.4 (.4) | 1.6 (.5) | 1.5 (.3) | 1.3 (.3) | 1.4 (0.4) |
| **Triglycerides (mmol/l)** | 1.3 (.7) | 1.3 (.8) | 1.2 (.7) | 1.2 (.9) | 1.2 (0.6) |
| **BMI (kg/m$^2$)** | 27.0 (4.3) | 27.2 (4.5) | 26.5 (3.8) | 26.4 (4.2) | 26.9 (4.6) |
| **Waist circumference (cm)** | 94.9 (12.8) | 95.2 (13.0) | 92.5 (11.7) | 92.5 (11.4) | 96.3 (11.7) |
| **Fasting glucose (mmol/l)** | 5.7 (1.0) | 5.5 (1.2) | 5.9 (.9) | 4.9 (.9) | 5.2 (1.4) |
| **Diabetes medication** | 4.1% | 5.3% | 4.4% | 0.2% | 12% |
| **Antihypertensive medication** | 19% | 21% | 22% | 8.1% | 60% |
| **Alcohol intake** | 6.9 (6.1) (g/week) | 6.9 (6.8) (g/week) | 2.43 (2.92) (drinks/week) | NA | NA |
| **Exercise habits** | 1.69 (1.38) (On a 6 grade scale) | 1.58 (1.43) (On a 6 grade scale) | 2.29 (.8) (On a 5 grade scale) | 2.8 (1.01) (On a 4 grade scale) | 1.21 (1.31) (On a 4 grade scale) |
| **Education** | | | | | |
| **<10 years** | 8% | 11% | 21% | 8% | 56% |
| **10–12 years** | 41% | 48% | 29% | 44% | 19% |
| **>12 years** | 51% | 41% | 50% | 48% | 25% |
| **Smokers** | 9.2% | 16% | 6.7 years of smoking | 9.8% | 3.2% |

NA = Not assessed

for traditional CVD risk factors (HR 1.32 for a 1 SD change, 95%CI 1.07–1.63, details in Table 3).

Of the 15 validated metabolites, we could only find published associated genetic variants (mQTL) with $p<5^*10^{-8}$ for glycine (rs715, position chr2:211543055:T/C, nearest gene *CPS1*). The Wald ratio for this loci vs MetS genetics was not significant (beta 0.11, SE 0.09, p-value 0.20). We did not evaluate if the metabolite glucose is causally related to MetS, since the glucose criteria is a part of MetS.

In the GWAS for MetS, 91 independent loci with $p<5^*10^{-8}$ were found [14]. When these loci were used as genetic instruments vs genetic data for glycine, glycerate or lactate (the only metabolites for which GWAS data were found), evidence was found for an association between shared genes between MetS and glycine (IVW beta 0.021, SE 0.007, p = 0.0027), but not for glycerate (IVW beta -0.0025, SE 0.006, p = 0.71), while lactate was of borderline significance (IVW beta 0.012, SE 0.006, p = 0.052).

## Discussion

The present study using several large samples identified 15 metabolites from a wide variety of metabolic pathways to be significantly related to all five components of the MetS. All of them were related to insulin sensitivity. One of these metabolites were related to incident CVD, 1-palmitoyl-2-oleoyl-GPE (16:0/18:1).

Previous studies have identified carnitine, 2-deoxyglucose, phenylalanine [19] hippurate [20], phosphatidylcholine 34:2, trimethylamine N-oxide (TMAO) [21, 22] glucose, aromatic

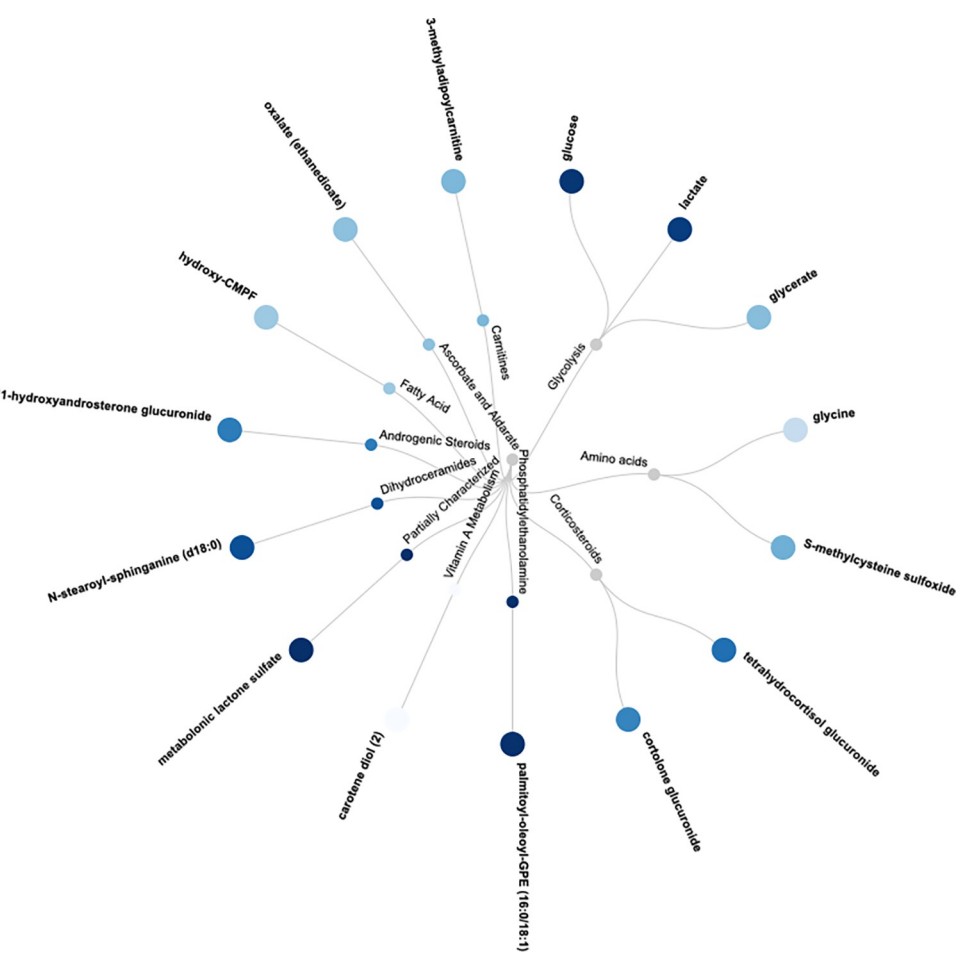

**Fig 1. Circular dendrogram of the 15 validated metabolites (and their chemical class) being related to all five metabolic syndrome (MetS) components.** A dark blue dot corresponds to a positive relationship vs MetS, while a light blue dot represents an inverse association. The details on the strengths of the relationships vs the MetS components are given in S2 Table, while the strengths of the relationships vs MetS (binary) are given in Table 2.

amino acids, salicyluric acid, maltitol, and p-cresol sulfate [23] to be linked to MetS. Using two independent samples, Roberts et al found 18 replicated metabolites to be linked to MetS, with representation from branched-chain amino acid metabolism, glutathione production, aromatic amino acid metabolism, gluconeogenesis, and the tricarboxylic acid cycle [24]. In another study [25], 16 metabolites, including carbohydrates, amino acids and several cholines was able to discriminate the MetS subjects vs controls with a C-statistic of 0.96. The novelty with the present study is that we identified metabolites related to all five components of the syndrome independent of BMI, with the hypothesis that some of these metabolites could be involved in pathways leading to the clustering of risk factors seen in some individuals.

All analyses in the present study were adjusted for BMI. This was done because obesity as such is related to each of the five MetS components, and we do not want to produce a long list of metabolites being related to all of the five MetS components just due to the fact that they are related to obesity. The fact that some metabolites are associated with both obesity (BMI) as such and fat distribution (WC) might be the reason why some obesity-related metabolites were related to all 5 risk factors despite adjustment for BMI.

Of interest is to note that the 15 metabolites found to be related to all five MetS components are involved in a variety of different metabolic classes, including amino acids, simple

**Table 2. Relationships between the 15 validated metabolites being related to all five components in the metabolic syndrome (MetS) and prevalent MetS in a meta-analysis of the PIVUS and POEM samples.** The results are sorted on p-value. A star following the p-value denotes that the metabolite shows a false discovery rate (FDR)<0.05.

| Super pathway | Sub pathway | Chemical name | Beta | 95%CI lower | 95%CI higher | p-value |
|---|---|---|---|---|---|---|
| Lipid | Phosphatidylethanolamine (PE) | 1-palmitoyl-2-oleoyl-GPE (16:0/18:1) | .57 | .38 | .78 | 1.12e-08* |
| Cofactors and Vitamins | Vitamin A Metabolism | carotene diol (2) | -.61 | -.83 | -.4 | 2.99e-08* |
| Partially Characterized Molecules | Partially Characterized Molecules | metabolonic lactone sulfate | .58 | .38 | .8 | 4.15e-08* |
| Carbohydrate | Glycolysis, Gluconeogenesis, and Pyruvate Metabolism | glucose | .55 | .35 | .76 | 8.39e-08* |
| Carbohydrate | Glycolysis, Gluconeogenesis, and Pyruvate Metabolism | lactate | .52 | .32 | .73 | 3.52e-07* |
| Lipid | Dihydroceramides | N-stearoyl-sphinganine (d18:0/18:0) | .43 | .23 | .64 | .000036* |
| Amino Acid | Glycine, Serine and Threonine Metabolism | glycine | -.33 | -.53 | -.14 | .00070* |
| Lipid | Corticosteroids | tetrahydrocortisol glucuronide | .29 | .09 | .49 | .0037* |
| Lipid | Androgenic Steroids | 11-beta-hydroxyandrosterone glucuronide | .23 | .04 | .42 | .018* |
| Lipid | Corticosteroids | cortolone glucuronide | .19 | -.01 | .41 | .058 |
| Lipid | Fatty Acid, Dicarboxylate | hydroxy-CMPF | -.17 | -.36 | .01 | .070 |
| Cofactors and Vitamins | Ascorbate and Aldarate Metabolism | oxalate (ethanedioate) | -.12 | -.33 | .08 | .22 |
| Carbohydrate | Glycolysis, Gluconeogenesis, and Pyruvate Metabolism | glycerate | -.099 | -.30 | .10 | .32 |
| Lipid | Fatty Acid Metabolism (Acyl Carnitine, Dicarboxylate) | pimeloylcarnitine/3-methyladipoylcarnitine (C7-DC) | -.067 | -.25 | .12 | .46 |
| Amino Acid | Methionine, Cysteine, SAM and Taurine Metabolism | S-methylcysteine sulfoxide | -.026 | -.22 | .16 | .77 |

carbohydrates, androgenic steroids and corticosteroids, ceramides, carnitines, and phospholipids. However, which of these metabolic classes that are involved in the pathogenesis of the clustering of risk factors could not be told from the present study.

**Table 3. Relationships between the 15 validated metabolites being related to all five components in the metabolic syndrome (MetS) and incident atherosclerotic cardiovascular disease in the EpiHealth cohort.**

| Metabolite | Age and sex-adjusted | | | | Multiple adjusted | | | |
|---|---|---|---|---|---|---|---|---|
| | HR | 95%CI low | 95%CI high | p-value | HR | 95%CI low | 95%CI high | p-value |
| 1-palmitoyl-2-oleoyl-GPE (16:0/18:1) | 1.43 | 1.17 | 1.75 | .00054 | 1.32 | 1.07 | 1.63 | .010 |
| N-stearoyl-sphinganine (d18:0/18:0) | 1.36 | 1.11 | 1.68 | .0032 | 1.23 | .98 | 1.57 | .074 |
| S-methylcysteine sulfoxide | .84 | .68 | 1.02 | .082 | .88 | .72 | 1.08 | .23 |
| carotene diol (2) | .84 | .69 | 1.04 | .10 | .92 | .73 | 1.15 | .45 |
| hydroxy-CMPF | .87 | .71 | 1.06 | .17 | .96 | .78 | 1.2 | .74 |
| tetrahydrocortisol glucuronide | 1.14 | .92 | 1.4 | .23 | 1 | .8 | 1.25 | .99 |
| oxalate (ethanedioate) | .89 | .71 | 1.09 | .24 | .98 | .79 | 1.22 | .84 |
| glycine | .89 | .72 | 1.11 | .28 | .96 | .76 | 1.21 | .72 |
| lactate | 1.12 | .91 | 1.36 | .28 | 1 | .81 | 1.25 | .97 |
| glycerate | .9 | .74 | 1.12 | .34 | .99 | .79 | 1.22 | .89 |
| metabolonic lactone sulfate | 1.09 | .89 | 1.34 | .39 | .94 | .75 | 1.19 | .62 |
| pimeloylcarnitine/3-methyladipoylcarnitine (C7-DC) | .94 | .77 | 1.15 | .55 | 1.02 | .83 | 1.27 | .82 |
| cortolone glucuronide (1) | 1.01 | .83 | 1.25 | .92 | .85 | .68 | 1.06 | .15 |
| 11beta-hydroxyandrosterone glucuronide | 1.01 | .83 | 1.23 | .92 | .94 | .76 | 1.16 | .59 |
| glucose | 1 | .82 | 1.22 | .98 | .86 | .68 | 1.07 | .18 |

MetS has been linked to the major cardiovascular diseases, myocardial infarction, stroke and heart failure [8]. MetS has also been linked to other adverse cardiovascular conditions, such as a poor outcome in patients affected by outflow tract premature ventricular contractions treated by catheter ablation [26] and a proarrythmogennic state in heart failure patients treated with an internal cardioverter defibrillator (ICD) [27]. An increased level of inflammation [28] together with over-stretch of cardiac muscle and fibrosis development due to MetS could lead to cardiac electrophysiological alterations, a poor myocardial performance and clinical outcome in the patients affected by MetS [29].

We found one metabolite, 1-palmitoyl-2-oleoyl-GPE (16:0/18:1), being related to all 5 MetS components in a validated fashion, to be associated with incidentCVD. Phosphatidyl-ethanolamines (GPE) are glycerophospholipids being mainly found on the inner part of the cell membrane and have been suggested to be involved in multiple actions, such as protein breakdown, mitochondrial function, autophagy and membrane fusion. One small case-control study of patients with lucunar brain infarcts showed GPE (35:2) to be increased in the cases [30]. Phosphatidylethanolamine (20:0/18:2) has been found to be reduced in subjects with severe coronary atherosclerosis [31]. It should however be acknowledged that the function of PEs might well be affected by the fatty acids included in the GPE. The fatty acids 16:0 and 18:1 included in 1-palmitoyl-2-oleoyl-GPE have both been associated with unwarranted health outcomes, such as insulin resistance, diabetes and myocardial infarction [32–34].

Already in the description of MetS in 1988 [1], insulin resistance was suggested to be the major mechanism behind the clustering of risk factors. It is therefore of interest to note that all of the 15 validated metabolites also were associated with insulin resistance as measured by the Matsuda index in a separate cohort following adjustment for BMI.

In light of the present findings that the metabolomic profile being in common for the five MetS components is very similar to that seen in insulin resistance, it would be of interest to evaluate the metabolomic profile in lean subjects with insulin resistance, as this group often suffer from NAFLD and have an increased risk of CVD [35, 36]. However, in the only study in which we have data on liver fat (the POEM study), only 3 subjects were classified as being lean and insulin resistant, so meaningful statistical evaluation of the metabolomic profile in this interesting group cannot be performed.

The major strength of the present study is the use of several independent large studies that made it possible to perform a discovery/validation approach with a good power despite the large number of metabolites evaluated. Another strength is that we had another two samples to test if the 15 metabolites also were linked to prevalent MetS and insulin resistance. The major weakness is that we ideally wanted to test the 15 metabolites vs incident MetS, but did not have a follow-up of future MetS in any of the samples. We did however have follow-up of incident cases of CVD, but since the number of cases were rather low, we did have a limited power to detect significant association, especially following multiple adjustment. This will possibly lead to false negative associations between some metabolites and incident CVD. We also acknowledge that we have been studying almost exclusively Swedish subjects with European descent and that our results have to be reproduced in other countries and in other ethnic groups. Another limitation is that we do not have access to strong genetic instruments for most metabolites, so the issue on causality could not be properly addressed by genetic studies. No formal power calculation was performed. The study is however the largest study (>11,000 individuals) performed regarding a large set of metabolites (n = 791) and MetS, so the power to find significant associations was substantially higher than in the previous literature.

In conclusion, a complex metabolic profile was disclosed being related to a clustering of the cardiovascular risk factors included in MetS. This profile was also related to the Matsuda index, further emphasizing the importance of insulin sensitivity as an important underlying

mechanism behind the clustering of risk factors. One of the metabolites been related to all MetS components, 1-palmitoyl-2-oleoyl-GPE (16:0/18:1), was also related to incident CVD, suggesting that this metabolite is worthwhile to explore in more detail as a potential mediator in the MetS vs atherosclerotic CVD relationship.

## Supporting information

**S1 Table. Overview of the metabolites being associated with at least one of the five MetS criteria.** The criteria are given together with sum of criteria. The table was sorted on sum of criteria.
(DOCX)

**S2 Table. Relationships between the 15 validated metabolites being related to all five components in the metabolic syndrome (MetS) and fasting glucose (GLU), HDL-cholesterol, systolic blood pressure (SBP), triglycerides (TG) and waist circumference (WC).** The estimates are from the validation step in the SCAPIS-Malmö cohort. No relationship for the metabolite glucose vs the GLU criteria is shown.
(DOCX)

## Author Contributions

**Conceptualization:** Lars Lind, Johan Ärnlöv.

**Investigation:** Lars Lind, Johan Sundström, Sölve Elmståhl, Koen F. Dekkers, J. Gustav Smith, Gunnar Engström, Tove Fall, Johan Ärnlöv.

**Writing – original draft:** Lars Lind.

**Writing – review & editing:** Johan Sundström, Sölve Elmståhl, Koen F. Dekkers, J. Gustav Smith, Gunnar Engström, Tove Fall, Johan Ärnlöv.

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
