## [Decision Letter · Decision Letter 0]

25 Jul 2022

PONE-D-22-17734THE METABOLOMIC PROFILE ASSOCIATED WITH CLUSTERING OF CARDIOVASCULAR RISK FACTORS – A MULTI-SAMPLE EVALUATIONPLOS ONE

Dear Dr. Lind,

Thank you for submitting your manuscript to PLOS ONE. After careful consideration, we feel that it has merit but does not fully meet PLOS ONE’s publication criteria as it currently stands. Therefore, we invite you to submit a revised version of the manuscript that addresses the points raised during the review process.

ACADEMIC EDITOR:The manuscript is interesting, but some issues need to be addressed  by authors.==============================

We look forward to receiving your revised manuscript.

Kind regards,

Ferdinando Carlo Sasso, PhD, MD

Academic Editor

PLOS ONE

Journal Requirements:

2. Please provide additional details regarding participant consent. In the ethics statement in the Methods and online submission information, please ensure that you have specified what type you obtained (for instance, written or verbal, and if verbal, how it was documented and witnessed). If your study included minors, state whether you obtained consent from parents or guardians. If the need for consent was waived by the ethics committee, please include this information. Please also state whether your data was analysed anonymously or if you had access to identifying information.

Additional Editor Comments:

The manuscript is interesting, but some issues need to be addressed by authors. Please, submit a revised version of your manuscript possibly within 2 weeks.

Reviewers' comments:

Reviewer's Responses to Questions

**Comments to the Author**

1. Is the manuscript technically sound, and do the data support the conclusions?

Reviewer #1: Yes

Reviewer #2: Yes

2. Has the statistical analysis been performed appropriately and rigorously? 

Reviewer #1: Yes

Reviewer #2: I Don't Know

3. Have the authors made all data underlying the findings in their manuscript fully available?

Reviewer #1: Yes

Reviewer #2: No

4. Is the manuscript presented in an intelligible fashion and written in standard English?

Reviewer #1: Yes

Reviewer #2: Yes

5. Review Comments to the Author

Reviewer #1: I read with great interest the paper “The metabolomic profile associated with clustering of cardiovascular risk factors – a multi-sample evaluation" by Lind et al.

The article is well written. Paper design must be improved. The article is logically divided into sections and subsections.

Comments:

1. Line 64-65: please better specify the criteria for metabolic syndrome diagnosis. High blood pressure or in treatment, low HDL, high triglycerides levels…

2. Line 79-81: I suggest the author to modify accordingly: “Metabolomics have extensively been used to characterize the metabolic landscape of obesity and diabetes [6, 7]. Moreover, several studies have also been published on metabolomics in MetS [8]”.

3. The key role of insulin resistance in metabolic syndrome have been stressed in various research, and it has been mostly associated with visceral adipose. However, newer reports have reported the presence of insulin resistance in lean individuals, with NAFLD development, as well as increased cardiovascular disease development (doi: 10.3390/antiox10020270; doi: 10.37349/emed.2020.00019). Pathophysiological mechanisms are still not clear. Though, the suggested metabolomics could be implicated in this process. Is there any evidence in such individuals? Do you have any data?

Reviewer #2: INTRODUCTION:

It is too long and not well focused on main study background, hypothesis, literature gap, and aim.

Please introduce the full diagnostic criteria for Metabolic Syndrome (MS), the criteria that you will use for all study. Add it in detail in the Methods.

Please add more information about MS and cardiovascular diseases (CVDs) genesis and prognosis. Indeed, authors showed that MS could favor an arrhythmic status leading to enhanced automatism and higher rate of arrhythmic events with consequent refractoriness to ablative approaches and worse clinical outcomes (Cardiovasc Disord. 2014 Dec 6;14:176. doi: 10.1186/1471-2261-14-176). Please discuss this point.

Again, the MS could cause over-inflammation and over-stretch of cardiac muscle (Front Physiol. 2018 Jun 26;9:758. doi: 10.3389/fphys.2018.00758), leading to worse prognosis by higher rate of arrhythmic atrial and ventricular events, ICDs’ therapies and hospitalizations. In this case, the MS could result in the compromising of functional status of heart failure patients treated with ICDs (Front Physiol. 2018 Jun 26;9:758. doi: 10.3389/fphys.2018.00758). Indeed, the MS could lead to cardiac electrophysiological alterations and clinical response in the treated patients affected by MS, by anormalities of sensing, (pacing) and impedance parameters (Medicine (Baltimore). 2017 Apr;96(14):e6558. doi: 10.1097/MD.0000000000006558). Please discuss this point and the adverse association between MS and HF.

METHODS:

Do you have number of ethical committee? Please add it.

How did you calculate the study sample size?

Please include a full description of incident atherosclerotic cardiovascular disease.

RESULTS:

I see a low percentage (0.2-12%) of diabetes medications. Please explain this point.

DISCUSSION:

Please focus the Discussion of 3 pages of description, and include according to authors (Curr Pharm Des. 2020;26(22):2565-2573. doi: 10.2174/1381612826666200213123029. ), the importance of over-inflammation in the pathogenesis and worse prognosis of CVDs, as in the case (see comments before) of MS. Please discuss it.

6. PLOS authors have the option to publish the peer review history of their article (what does this mean?). If published, this will include your full peer review and any attached files.

Reviewer #1: No

Reviewer #2: No

---

## [Author Response · Author response to Decision Letter 0]

29 Aug 2022

Reviewer #1: I read with great interest the paper “The metabolomic profile associated with clustering of cardiovascular risk factors – a multi-sample evaluation" by Lind et al.

The article is well written. Paper design must be improved. The article is logically divided into sections and subsections.

Comments:

1. Line 64-65: please better specify the criteria for metabolic syndrome diagnosis. High blood pressure or in treatment, low HDL, high triglycerides levels…

Reply: We have now better specified the criteria for metabolic syndrome diagnosis: “high blood pressure, increased waist circumference, high fasting glucose, low HDL-cholesterol and increased triglycerides.” (line 63-64)

In addition, we have given the details used to define the 5 components in the methods section: high blood pressure, increased waist circumference, high fasting glucose, low HDL-cholesterol and increased triglycerides (line 139-143) “The five components were defined as follows: Blood pressure ≥ 130/85 mmHg or antihypertensive treatment, fasting plasma glucose ≥ 6.1 mmol/l or antidiabetic treatment, serum triglycerides ≥ 1.7 mmol/l, waist circumference > 102 cm in men and > 88 cm in women, HDL-cholesterol < 1.0 mmol/l in men and < 1.3 in women. Three of the mentioned five criteria should be fulfilled for MetS.”

2. Line 79-81: I suggest the author to modify accordingly: “Metabolomics have extensively been used to characterize the metabolic landscape of obesity and diabetes [6, 7]. Moreover, several studies have also been published on metabolomics in MetS [8]”.

Reply: This part has now been changed according to your suggestion (line 74-76).

3. The key role of insulin resistance in metabolic syndrome have been stressed in various research, and it has been mostly associated with visceral adipose. However, newer reports have reported the presence of insulin resistance in lean individuals, with NAFLD development, as well as increased cardiovascular disease development (doi: 10.3390/antiox10020270; doi: 10.37349/emed.2020.00019). Pathophysiological mechanisms are still not clear. Though, the suggested metabolomics could be implicated in this process. Is there any evidence in such individuals? Do you have any data?

Reply: This is a good idea! This lean insulin resistant group with NAFLD is very interesting. However, in the only study in which we have data on liver fat (the POEM study), only 3 subjects were classified as being lean and insulin resistant, so meaningful statistical evaluation of the metabolomic profile in this interesting group cannot be performed. We have now added a para on this idea in the discussion section, including the references you suggested (line 339-345): “In light of the present findings that the metabolomic profile being in common for the five MetS components is very similar to that seen in insulin resistance, it would be of interest to evaluate the metabolomic profile in lean subjects with insulin resistance, as this group often suffer from NAFLD and have an increased risk of CVD [35,36]. However, in the only study in which we have data on liver fat (the POEM study), only 3 subjects were classified as being lean and insulin resistant, so meaningful statistical evaluation of the metabolomic profile in this interesting group cannot be performed.”

Reviewer #2: INTRODUCTION:

It is too long and not well focused on main study background, hypothesis, literature gap, and aim.

Reply: We have now shortened the introduction and tried to be more focused on the items you suggest.

Please introduce the full diagnostic criteria for Metabolic Syndrome (MS), the criteria that you will use for all study. Add it in detail in the Methods.

Reply: We have now given the details used to define the 5 components in the methods section: high blood pressure, increased waist circumference, high fasting glucose, low HDL-cholesterol and increased triglycerides (line 139-143) “The five components were defined as follows: Blood pressure ≥ 130/85 mmHg or antihypertensive treatment, fasting plasma glucose ≥ 6.1 mmol/l or antidiabetic treatment, serum triglycerides ≥ 1.7 mmol/l, waist circumference > 102 cm in men and > 88 cm in women, HDL-cholesterol < 1.0 mmol/l in men and < 1.3 in women. Three of the mentioned five criteria should be fulfilled for MetS.”

Please add more information about MS and cardiovascular diseases (CVDs) genesis and prognosis. Indeed, authors showed that MS could favor an arrhythmic status leading to enhanced automatism and higher rate of arrhythmic events with consequent refractoriness to ablative approaches and worse clinical outcomes (Cardiovasc Disord. 2014 Dec 6;14:176. doi: 10.1186/1471-2261-14-176). Please discuss this point.

Again, the MS could cause over-inflammation and over-stretch of cardiac muscle (Front Physiol. 2018 Jun 26;9:758. doi: 10.3389/fphys.2018.00758), leading to worse prognosis by higher rate of arrhythmic atrial and ventricular events, ICDs’ therapies and hospitalizations. In this case, the MS could result in the compromising of functional status of heart failure patients treated with ICDs (Front Physiol. 2018 Jun 26;9:758. doi: 10.3389/fphys.2018.00758). Indeed, the MS could lead to cardiac electrophysiological alterations and clinical response in the treated patients affected by MS, by anormalities of sensing, (pacing) and impedance parameters (Medicine (Baltimore). 2017 Apr;96(14):e6558. doi: 10.1097/MD.0000000000006558). Please discuss this point and the adverse association between MS and HF.

Reply: We have included a new paragraph in the discussion section discussing the issues of hyperinflammation, ICD, cardiac stretch and heart failure in relation to MetS with three new references mentioned by you. However, since you wanted to reduce the discussion part to three pages, this added paragraph had to be kept short (line 315-322):” MetS has been linked to the major cardiovascular diseases, myocardial infarction, stroke and heart failure [9]. MetS has also been linked to other adverse cardiovascular conditions, such as a poor outcome in patients affected by outflow tract premature ventricular contractions treated by catheter ablation [26] and a proarrythmogennic state in heart failure patients treated with an internal cardioverter defibrillator (ICD) [27]. An increased level of inflammation [28] together with over-stretch of cardiac muscle and fibrosis development due to MetS could lead to cardiac electrophysiological alterations, a poor myocardial performance and clinical outcome in the patients affected by MetS [29].”

METHODS:

Do you have number of ethical committee? Please add it.

Reply: This has now been added (line 102): “Dnr 2021-00134”

How did you calculate the study sample size?

Reply: No formal power calculation was performed. The study is however the largest study (>11,000 individuals) performed regarding a large set of metabolites (n=791) and MetS, so the power to find significant association is higher than in the previous literature. We have now added to the discussion (line 359-362):” No formal power calculation was performed. The study is however the largest study (>11000 individuals) performed regarding a large set of metabolites (n=791) and MetS, so the power to find significant associations was substantially higher than in the previous literature.”

Please include a full description of incident atherosclerotic cardiovascular disease.

Reply: We have now added more information regarding the evaluation of incident atherosclerotic cardiovascular disease. If you like us to add some additional information, please let us know.” Using data from the Swedish cause of death and in-hospital care registers, we defined a combined end-point for atherosclerotic CVD being either fatal or non-fatal acute myocardial infarction or ischemic stroke (ICD-10 codes I20 or I63-I66). Incident cases of atherosclerotic CVD were only investigated in the EpiHealth sample, since the other samples had yet too short follow-up period. The median follow-up period in EpiHealth was 8.6 years. The censor date of the follow-up was Dec 31, 2020.” (line 166-171)

RESULTS:

I see a low percentage (0.2-12%) of diabetes medications. Please explain this point.

Reply: This wide range in diabetes prevalence is due to the fact that the range in age is large between the samples. The diabetes prevalence in the different age-groups are similar to other Swedish cohort studies, so from our perspective the percentage (0.2-12%) of diabetes medications is not surprising.

DISCUSSION:

Please focus the Discussion of 3 pages of description, and include according to authors (Curr Pharm Des. 2020;26(22):2565-2573. doi: 10.2174/1381612826666200213123029. ), the importance of over-inflammation in the pathogenesis and worse prognosis of CVDs, as in the case (see comments before) of MS. Please discuss it.

Reply: We have now condensed the discussion to three pages. We have now discussed the proinflammatory state as a driver of CVD in subjects with MetS, as commented before.

---

## [Decision Letter · Decision Letter 1]

2 Sep 2022

The metabolomic profile associated with clustering of cardiovascular risk factors – a multi-sample evaluation

PONE-D-22-17734R1

Dear Dr. Lind,

We’re pleased to inform you that your manuscript has been judged scientifically suitable for publication and will be formally accepted for publication once it meets all outstanding technical requirements.

Kind regards,

Ferdinando Carlo Sasso, PhD, MD

Academic Editor

PLOS ONE

Additional Editor Comments (optional):

The authors addressed all issues raised by reviewers. No further comments.

Reviewers' comments:

Reviewer's Responses to Questions

**Comments to the Author**

1. If the authors have adequately addressed your comments raised in a previous round of review and you feel that this manuscript is now acceptable for publication, you may indicate that here to bypass the “Comments to the Author” section, enter your conflict of interest statement in the “Confidential to Editor” section, and submit your "Accept" recommendation.

Reviewer #1: All comments have been addressed

Reviewer #2: All comments have been addressed

2. Is the manuscript technically sound, and do the data support the conclusions?

Reviewer #1: Yes

Reviewer #2: Yes

3. Has the statistical analysis been performed appropriately and rigorously? 

Reviewer #1: Yes

Reviewer #2: Yes

4. Have the authors made all data underlying the findings in their manuscript fully available?

Reviewer #1: Yes

Reviewer #2: Yes

5. Is the manuscript presented in an intelligible fashion and written in standard English?

Reviewer #1: Yes

Reviewer #2: Yes

6. Review Comments to the Author

Reviewer #1: The paper has much improved and the authors managed to respond to all the issues I raised. The paper can be further processed for publication.

Reviewer #2: The authors revised the article according to reviewers' comments.

In my opinion, you could be accepted for a possible pubblication in the journal.

7. PLOS authors have the option to publish the peer review history of their article (what does this mean?). If published, this will include your full peer review and any attached files.

Reviewer #1: No

Reviewer #2: No

---

## [Editor Report · Acceptance letter]

6 Sep 2022

PONE-D-22-17734R1 

The metabolomic profile associated with clustering of cardiovascular risk factors – a multi-sample evaluation 

Dear Dr. Lind:

I'm pleased to inform you that your manuscript has been deemed suitable for publication in PLOS ONE. Congratulations! Your manuscript is now with our production department. 

Kind regards, 

on behalf of

Professor Ferdinando Carlo Sasso 

Academic Editor

PLOS ONE